The use of wireless sensors in the neonatal intensive care unit: a study protocol

Senechal Eva 1
Radeschi Daniel 2
Tao Lydia 3
Lv Shasha 3
Jeanne Emily 1
Kearney Robert 2
Shalish Wissam 1 3
Sant Anna Guilherme 1 3 guilherme.santanna@mcgill.ca
1 Department of Experimental Medicine, McGill University , Montréal, Quebec , Canada
2 Department of Biomedical Engineering, McGill University , Montréal, Quebec , Canada
3 Department of Pediatrics, McGill University Health Center , Montréal, Quebec , Canada
Fujioka Kazumichi
Electronic publication date: 2023 Jun 27
Publication date: 2023
Volume: 11
Electronic Location ID: e15578
Received 2023 Feb 9; Accepted 2023 May 25
Copyright: © 2023 Senechal et al.
Copyright year: 2023
Copyright holder: Senechal et al.
License: This is an open access article distributed under the terms of the Creative Commons Attribution License, which permits unrestricted use, distribution, reproduction and adaptation in any medium and for any purpose provided that it is properly attributed. For attribution, the original author(s), title, publication source (PeerJ) and either DOI or URL of the article must be cited.
License URL: https://creativecommons.org/licenses/by/4.0/

Keywords: Wireless monitoring, Wearable technology, NICU, Patient monitoring, Pediatric care, PICU

Funding: Montreal Children’s Hospital Foundation, under the Smart Hospital’s Project This research is funded by the Montreal Children’s Hospital Foundation, under the Smart Hospital’s Project. The funders had no role in study design, data collection and analysis, decision to publish, or preparation of the manuscript.

==============================
Background

Continuous monitoring of vital signs and other biological signals in the Neonatal Intensive Care Unit (NICU) requires sensors connected to the bedside monitors by wires and cables. This monitoring system presents challenges such as risks for skin damage or infection, possibility of tangling around the patient body, or damage of the wires, which may complicate routine care. Furthermore, the presence of cables and wires can act as a barrier for parent-infant interactions and skin to skin contact. This study will investigate the use of a new wireless sensor for routine vital monitoring in the NICU.

Methods

Forty-eight neonates will be recruited from the Montreal Children’s Hospital NICU. The primary outcome is to evaluate the feasibility, safety, and accuracy of a wireless monitoring technology called ANNE® One (Sibel Health, Niles, MI, USA). The study will be conducted in 2 phases where physiological signals will be acquired from the standard monitoring system and the new wireless monitoring system simultaneously. In phase 1, participants will be monitored for 8 h, on four consecutive days, and the following signals will be obtained: heart rate, respiratory rate, oxygen saturation and skin temperature. In phase 2, the same signals will be recorded, but for a period of 96 consecutive hours. Safety and feasibility of the wireless devices will be assessed. Analyses of device accuracy and performance will be accomplished offline by the biomedical engineering team.

Conclusion

This study will evaluate feasibility, safety, and accuracy of a new wireless monitoring technology in neonates treated in the NICU.

Introduction

Background

In Canada, approximately 11.1% of newborns require admission to the Neonatal Intensive Care Unit (NICU) after birth (Fallah et al., 2011). Common causes for admission are prematurity, low birth weight, respiratory distress syndrome, apnea of prematurity, and hypoglycemia (Gallacher, Hart & Kotecha, 2016). NICU patients require around the clock care and continuous monitoring of their vital signs including heart rate (HR), temperature, oxygen saturation (SpO2), and respiratory rate (RR). Monitoring of these signs is essential for providing effective care.

Current standard monitoring technologies involve sensors that are applied to the body and connected to bedside monitors using wires and cables. Heart rate is measured via small electrodes connected via wires and cables to the bedside monitor that continuously displays both electrocardiogram (ECG) tracings and average HR. Temperature is monitored via a probe placed at the axilla and oxygen saturation (SpO2) via a sensor placed on the extremities (fingers, hand, wrists, or foot) with the values and signal tracing displayed on the bedside monitor. Finally, respiratory rate is measured via thoracic impedance using the same sensors and wires as the ECG (Kumar et al., 2019). Thus, current standard of care is a wired monitoring system involving multiple sensors applied to the skin with long wires and cables connected to the bedside monitors.

The presence of multiple wires poses several hindrances. First, wires can act as a barrier to parent-infant bonding (Al Maghaireh et al., 2016; Jämsä & Jämsä, 1998; Lantz & Ottosson, 2012). In a systematic review, parents expressed that the complex wired system in their infant’s crib made them reluctant to hold the baby. Bonding is important for the parent’s emotional health and is also associated with better outcomes for the infants (Mehrpisheh et al., 2022). Indeed, skin-to-skin contact or Kangaroo-care, has been linked to pain reduction, and better physiological and behavioral outcomes (Conde-Agudelo, Belizán & Diaz-Rossello, 2011). Second, the multiple wires can tangle and restrict the baby’s natural movement leading to discomfort or touch the floor, increasing the risk of nosocomial infections (Bonner et al., 2017; Russotto et al., 2015). To avoid tangling and movement restriction, sensors are often removed and reapplied which can cause harm to the sensitive neonatal skin and disturb infants’ rest, both contributing to pain and distress (Chen et al., 2009). Nurses reported that any technology able to reduce handling of premature babies would significantly improve patient outcomes (Bonner et al., 2017).

Interestingly, monitoring technology in neonatal care has undergone little change in the last five decades since the addition of pulse oximetry as standard of care (Xu et al., 2021). Despite witnessing rapid technological advances in smart monitoring and wearable devices across industries, the hospital sector has lagged in implementing these devices (Lewy, 2015; De Georgia et al., 2015). However, in recent years there have been interesting developments on wireless and wearable technologies for infant monitoring. Some consumer technologies such as the Owlet Smart Sock, MonBaby, and Lumi have been designed for neonates but are not FDA-cleared Class III medical devices (Al Maghaireh et al., 2016; Hewitt et al., 2019; Roh, 2021). While these devices have the potential for hospital monitoring, they must meet rigorous criteria for medical devices, and preliminary studies have shown limitations in their accuracy (Bonafide et al., 2018; Bonafide, Jamison & Foglia, 2017).

Recently, a vital sign monitoring system called ANNE® Monitoring System was developed that uses small, light wireless sensors attached to the newborn’s skin (Chung et al., 2019). A pilot project including 21 neonates with gestational age between 28 weeks to full term, demonstrated good safety and efficacy in detecting heart rate, respiratory rate, temperature, and SpO2 when compared to current standards of care, over a 5-min period. In a second study, investigators strengthened the mechanical properties of the sensors, and examined different power-support schemes, including modular primary batteries, embedded secondary batteries, and wireless power transfer (Chung et al., 2020). Additional features such as the continuous capture of body orientation and movements and acoustic signals of respiratory and cardiac activity via tri-axial accelerometers were added. This updated version was tested on 50 NICU & Pediatric Intensive Care Unit (PICU) patients, ranging from extremely preterm to term patients. Results showed that all signals obtained from wireless technologies were within FDA requirements for agreement with the standard measures; there were no negative skin reactions; and heat generation was negligible. However, the strength of this analysis was limited by the small sample sizes. Comparative analysis of signals was only reported for a select number of patients. In the case of HR, SpO2, and temperature, comparative analysis was only reported in three patients, while for RR, comparative analysis was only reported in six patients. Skin condition and heat generation was assessed and reported for all patients. In both preliminary studies the investigators demonstrated reliable, safe wireless data transmission, real-time signal processing and data display. Therefore, in this study we will test the feasibility, safety, and accuracy of ANNE® One for longer periods of monitoring in a larger number of neonates, of variable gestational ages and clinical conditions, receiving care in the NICU.

Study objectives

The objectives of the study are to: (1) demonstrate the feasibility of continuous wireless monitoring in term and preterm infants with variable degrees of maturation and clinical stability in the NICU; (2) assess safety of using a special wireless monitoring system in neonates; and (3) evaluate the accuracy, signal coverage and quality of the proposed wireless technology as compared to standard monitoring technology in the NICU.

Materials and Methods

Study design

This is a single center prospective observational study aimed at determining the safety, feasibility, and accuracy of wireless monitoring in the Neonatal Intensive Care Unit. This study design conforms to recommendations by the Standard Protocol Items: Recommendations for Interventional Trials (SPIRIT) (see Additional File 1).

Study setting

This study will be conducted in the NICU at the Montreal Children’s Hospital.

Research ethics approval

This study has been reviewed and approved by the Pediatric (PED) panel of the Research Ethics Board of MUHC (#2022-7704). The study was also registered on clinicaltrials.gov on July 8, 2021 (NCT04956354). Any changes in protocol modifications will be communicated to the Research Ethics Board of the MUHC by the principal investigators.

Consent to participate

All participants will provide article consent which will be obtained by member of research team prior to taking part in the study. Any modifications to the protocol which may impact the study’s intervention, target enrollment group, and outcomes will result in a formal amendment to the protocol. All amendments will be submitted and reviewed by the Research Ethics Board for approval and reflected in the registration on clinicaltrials.gov.

Eligibility criteria

Three different age groups will be included: Group 1 = term infants (37+0 to 40+6 weeks); Group 2 = preterm infants with postmenstrual ages 28+0 to 36+6 weeks, and Group 3 = extremely preterm infants with postmenstrual ages 23+0 to 27+6 weeks. Groups were designed by neonatologist to reflect a variety of common clinical situations encountered in the NICU. Each age group will consist of a variety of subgroups based on different clinical criteria for inclusion (Table 1). Infants with congenital anomalies and surgical conditions (gastroschisis, omphalocele, congenital diaphragmatic hernia), congenital heart disorders, will be excluded to reduce interference of research with care, and reduce interruptions in data collection. Additionally, infants congenital skin infections or known conditions with fragile skin (such as epidermolysis bullosa) will be excluded to prevent additional discomfort or complications associated with adhesives. See Fig. 1 for enrollment plan.

Table 1 Study population characteristics.

Groups	Clinical characteristics	Phase 1
(n = 24)	Phase 2
(n = 24)	
Group 1—Term infants with postmenstrual ages at enrollment of 37–42 weeks’	A: healthy* infants in room air at enrollment (N = 6)	3	3	
B: infants with perinatal asphyxia undergoing therapeutic hypothermia at enrollment (N = 6)	3	3	
Group 2—Preterm infants with postmenstrual ages at enrollment of 28+0 to 36+6 weeks	C: healthy* infants in room air at enrollment (N = 6)	3	3	
D: infants on continuous positive airway pressure at enrollment (N = 6)	3	3	
Group 3—Preterm infants with postmenstrual ages at enrollment of 23+0 to 27+6 weeks	E: infants on conventional mechanical ventilation at enrollment (N = 6)	3	3	
F: infants on high frequency ventilation at enrollment (N = 6)	3	3	
G: infants on nasal intermittent positive end expiratory pressure at enrollment (N = 6)	3	3	
H: infants on continuous positive airway pressure at enrollment (N = 6)	3	3	
Total:		24	24	
Note:

* Healthy in this study is defined as babies in room air, not requiring respiratory support such as high flow nasal canula (HFNC), continuous positive airway pressure (CPAP), Nasal intermittent positive pressure ventilation (NIPPV), conventional mechanical ventilation (CMV), no evidence of major congenital anomalies, neurological problems, necrotizing enterocolitis (NEC) or infection. The reason baby is admitted in the NICU will be recorded.

Figure 1 Study enrollment flow diagram.

Flow chart describing the flow of patient recruitment for this study.

Interventions

Study equipment

This project will use the ANNE® One from Sibel Health (Fig. 2). ANNE® is a wireless vital sign monitoring system that uses soft, flexible, skin-mountable biosensors (ANNE® Chest and ANNE® Limb) enabled with Bluetooth® 5 encrypted data communication to a mobile device application. It consists of two units which together obtain heart rate, respiratory rate, temperature, oxygen saturation, and movement data. The chest unit is secured via a hydrogel adhesive, and the limb unit via a fabric strap. These two units are rechargeable, and on full charge have a 36-h battery life. Devices take approximately 3 h to fully recharge.

Figure 2 ANNE® one wireless monitoring system.

All components provided with ANNE® One monitoring system, Image provided by Sibel Inc., Illinois, USA and approved for publication.

The device is currently under FDA review, and in collaboration with Sibel Health, an Investigational Testing Authorization (ITA) was obtained with Health Canada to test the monitoring system in this study.

The standard monitoring technology used as a reference will be the Philips Intellivue monitor (Intellivue MX450; Philips Healthcare, Best, Netherlands) connected to ECG leads (Neotrode®; ConMed, New York, NY, USA) and pulse oximeter probe (Red 24 LNC-10; Masimo, Irvine, CA, US).

The study will be performed in two phases. In phase 1, infants will undergo monitoring for a period of 8 h per day during daytime (from 8 am to 4 pm or from 9 am to 5 pm) and for 4 consecutive days. In phase 2, another group of infants selected based on the same postmenstrual age and clinical criteria as used in phase 1, will undergo monitoring for a period of 96 h continuously (Table 1). The study was broken down into two phases to allow the research team to be on site for all preliminary monitoring sessions in phase I, and thus facilitate the evolution to phase II which will imitate real continuous monitoring conditions, including overnight monitoring, without the presence of study personnel. Participation in the study will have no impact on routine patient care, and sensors will be removed and reapplied when necessary for care. Only in the event of serious adverse reaction parental withdrawal of consent, or physician request will the patient study be discontinued.

In phase 1, the study equipment will be placed into the patient room and two wireless sensors will be applied to the infant’s skin, one at the chest and the other one at the limb. In phase 2, an additional wireless sensor will be applied at the abdominal wall for acquisition of abdominal movements. The types of sensors to be used in each phase are shown in Table 2 and illustrated in Fig. 3.

Table 2 Sensors to be applied on infants in Phase 1 and 2.

Phase	Sensor	
Phase 1	Sensor 1, Sensor 3	
Phase 2	Sensor 1, Sensor 2, Sensor 3	

Figure 3 ANNE® one wireless sensors.

Two devices comprising ANNE® One monitoring system and signals to be acquired by the chest and limb sensors. ANNE® One chest obtains electrocardiography, accelerometry, and temperature. ANNE® One limb: obtains photoplethysmography, temperature. Image provided and approved for publication by Sibel Inc. (Chicago, IL, USA).

Sensor 1—A chest unit (ANNE® Chest) with embedded battery, ECG electrodes, clinical grade thermometer, and a three-axis accelerometer that captures the following signals: electrocardiography (ECG), temperature, and chest wall movements. The sensor will be placed in the middle of the infant’s chest.

Sensor 2 (phases 1 and 2)—A limb unit (ANNE® Limb) with embedded battery, clinical grade thermometer, and two LEDs along with a photodiode that capture photoplethysmogram (PPG) at each wavelength of light, from which the SpO2 is calculated. The sensor will be placed on either ankle-to-base of the foot or the wrist-to-hand, with rotations done as per standard of care (every 4 h for preterm infants and every 12 h for term infants).

Sensor 3 (phase 2 only) An abdominal unit (ANNE® Chest) with embedded battery, sensors, and a three-axis accelerometer that captures the following signals: electrocardiography (ECG), temperature, and abdominal wall movements. Only signals of abdominal wall movements will be sampled. The sensor will be placed above the umbilical line.

During the monitoring periods a Biosensors Data Aggregation and Synchronization (BioDASh) system developed by iKinesia Inc (Montreal, Quebec, Canada) for this study will simultaneously record the vital signals from both systems: wired and wireless. The only vital sign recording that will not be recorded continuously by BioDash is the standard temperature which is recorded using a clinical grade handheld thermometer or by a temperature sensor adhesive connected to the incubator. The reasons for this are: (a) temperature is not continuously recorded in all patients in the NICU and (b) BioDash is not able to connect and record signals from to the incubator.

The BioDash application will be running on the study laptop attached to the study cart in the patient room and display all signals in real time, store them, as well as allow team to make live annotations to data. A list of standardized annotations was created to add more details to recorded data. The full list of standardized annotations is provided in Table S1. Additionally, BioDash’s has a visual display providing information about serial number of paired sensors, battery level of the wireless sensors, electrodes contact with the skin, and strength of connection between sensors and the tablet.

In addition, photographs of the skin at the site of sensors placement will be taken before placement and after removal.

Study phases

Phase 1: Clinical and physiological data will be continuously acquired for 8 h during daytime, for 4 consecutive days. The wireless sensors will be visually assessed and disinfected using 70% isopropyl wipes prior to placement and following device removal each day of the study. Sensors will be disinfected using the same solution at the end of the study.

The ECG and SpO2 leads will be connected to a Philips Intellivue monitor (Intellivue MX450; Philips Healthcare, Best, Netherlands) to capture ECG and HR, PPG, SpO2, and respiratory waveform and rate data to be displayed and exported in real-time using a direct cable connected to a personal laptop computer via MediCollector software (Boston, MA, USA). At the beginning of each day, ANNE® sensors are selected for pairing by the researcher with the BioDash recording system based on serial number displayed both on tablet and devices. Data from the paired ANNE® sensors will be transferred via Bluetooth to a study tablet which will simultaneously send this data to the study laptop. The following waveforms and vital signs from both systems (via Philips Intellievue MX450) and ANNE® sensors will be recorded: ECG and heart rate, PPG, SpO2, respiratory waveform and respiratory rate signals. In addition, body temperature will be recorded from ANNE® sensors only. The BioDASh application running on the study laptop will stream, display, and log data in real-time. In addition, photographs of the skin at the site of placement of the sensors will be taken at baseline, time of device removal.

Phase 2: Clinical and physiological data will be acquired for 96 consecutive hours. The sensors will be replaced every 24 h or whenever low battery is detected. The wireless sensors will be visually assessed and disinfected using 70% isopropyl wipes prior to initial device placement, at each replacement, and following device removal at the end of the study. The same signals as in Phase I will be recorded by the BioDash Application.

In phase 2, respiration will also be recorded using uncalibrated respiratory inductive plethysmography (RIP—Respitrace QDC®; Viasys® Healthcare, Conshohocken, PA, USA), with the belts applied at the level of the chest and abdomen, for a period of 2 to 3 continuous hours each day, to measure chest and abdominal wall movements, respectively. The Respitrac® signals will also be displayed in real-time on the BioDASh application using the National Instruments (NI) data acquisition system (National Instruments, Austin, TX, USA) via Bayonet Neil-Concelman (BNC) cables to an NI (National Instruments, Austin, TX, USA) data acquisition system (NI USB-6212 BNC). Respitrace® signals will be displayed and logged in real-time on the study laptop by the BioDASh application. Acquired signals from all three devices will be stored by BioDASh and used for offline analysis. In addition, photographs of the skin at the site of placement of the sensors will be taken at baseline, time of device replacement, and 96 h.

Outcomes

Objective 1. Demonstrate the feasibility of continuous wireless monitoring in term and preterm infants with variable degrees of maturation and acuity in the NICU

Feasibility will be assessed by reporting challenges associated with sensor placement and removal, and acceptability of the continuous wireless monitoring system by parents and bedside nurses.

Data Annotations: To assess the feasibility of using these wireless sensors the number of times a sensor was readjusted, removed, or replaced will be collected via the live annotations added in BioDash by the researchers. Researchers may also add annotations related to difficulties applying and removing the sensors.

Parent and Nurse Survey: The research team will also approach parents and bedside nurses at the end of the 4 days with a short, anonymous survey to obtain perceptions about the wireless wearable devices (Additional File 2). The survey consists of three questions: Were you satisfied with the wireless sensors placed on your baby during their participation in this project? (Score 0—not satisfied at all to 10—very satisfied)

What are things you liked about the wireless sensors?

Did you encounter any problems with the wireless sensors? If yes, please specify.

Objective 2. Assess safety of using a special wireless monitoring system in neonates

Safety will be assessed by reporting any adverse skin reactions to sensors as well as evaluating pain at the time of device removal.

Skin Evaluation: Prior to device placement and following removal, photographs of the skin at the site of device placement will be taken. A board-certified dermatologist will evaluate these pictures and score according to the Neonatal Skin Condition Score (NSCS). The NSCS has been widely validated for assessment of skin conditions in neonates (Lund & Osborne, 2004) and consists of three items graded from 0 to 3 with final scores ranging from 3 (perfect) to 9 (worse).

Neonatal Infant Pain Scale: Upon sensors removal, the researcher will complete the Neonatal Infant Pain Scale (NIPS) to ensure that removal of adhesive does not cause discomfort to the patient. This scale has been widely validated for the assessment of acute pain in neonates (Witt et al., 2016) and consists of six items: facial expressions, cry, breathing patters, positioning of arms and legs, and state of arousal. All items except cry, which is scored from 0 to 2, are scored from 0 to 1 based. Scores range from 0 to 7, with scores greater than 3 indicating pain.

Objective 3. Evaluate the accuracy and coverage of the proposed wireless technology as compared to standard monitoring technology in the NICU

Accuracy will be assessed by comparing both the contents and the quality of the data obtained by the ANNE® sensors and the standard monitoring technology.

In terms of the contents, the recorded data of the raw heart rate (HR), temperature, oxygen saturation (SpO2), and respiratory rate (RR) obtained by the ANNE® sensors will be compared to the raw signals obtained by the standard monitoring system. A detailed list of the statistics that will be obtained from the data for comparison can be found in Table 3.

Table 3 List of desired test statistics derived from raw monitor signals.

Heart rate	Average heart rate per epoch (via histogram); number of bradycardias and tachycardia per hour; average and maximal duration of bradycardias (<100 beats/min) and tachycardia (>160) per recording (8-h epoch in phase I, 24-h epoch in phase II); number of bradycardias and tachycardia >10 s per hour or per epoch; average heart rate per epoch (via tachogram).	
ECG (time domain metrics)	minimum, maximum, and median NN interval (i.e., the interval between successive R-waves); standard deviation of the NN intervals (SDNN); the standard deviation of the averages of NN intervals in all 5-min segments of the recording (SDANN); the mean of the standard deviations of all NN intervals for all 5-min segments of the recording (SDNNi); the percentage of adjacent NN intervals differing by >50 ms (pNN50); root mean square of successive differences of NN intervals (RMSSD); total number of all NN intervals divided by the height of the 32-bin histogram of all NN intervals (triangular index; HRVTi); and the baseline width of the minimum square difference triangular interpolation of the highest peak of the 32-bin histogram of all NN intervals (TINN).	
ECG (frequency domain metrics)	TP: total power (<0.4 Hz), VLF: very low frequency (<0.04 Hz), LF: low frequency (0.04–0.15 Hz), HF: high frequency (>0.15 to <0.4 Hz), and LF/HF ratio using the Welch periodogram	
Oxygen saturation	Average, median and interquartile range of SpO2 and percentiles (5th and 95th); percentage time with values in 5% groupings between 70% and 100%; percentage times spent with SpO2 below 80% and above 95%; values in 5% groupings between 70% and 100%	
Respiratory rate	Average value (via histogram), number of bradypnea (<40 breaths/min) and tachypnea (>60 breaths/min) per hour; average and maximal duration of bradypnea and tachypnea per recording (8-h epoch in phase I, 24-h epoch in phase II); number of bradypnea and tachypnea >10 s per hour or per epoch; standard deviation, correlation, …tachypnea (above 60), bradypnea (below 40)	
Temperature	Average temperature; measurements of temperature variability (using standard deviation of the daily average temperature); percentage times spent in hypothermia (temperature <36.0 °C) and hyperthermia (temperature > 37.5 °C)	
Note:

SDNN, SD of normal- to- normal (NN) or R-wave- to- R-wave intervals; SDANN, SD of the averages of NN intervals in all 5 min segments of the entire recording; SDNNi, mean of the SDs of all NN intervals; RMSSD, root mean square of the differences between adjacent NN intervals; (f) SDSD, SD of differences between adjacent NN intervals; TINN, baseline width of the minimum square difference triangular interpolation of the highest peak of the histogram of all NN intervals measured.

The quality of the signal will be assessed by quantifying the average noise level for each signal, for both monitoring devices.

Continuity of signal will be assessed by quantifying the frequency and duration of connection loss, corruption of signal due movement artifacts, as well as quantification of portion of usable data from each recording period.

Participant timeline

The overall planned duration of the study is 24 months. Initiation of enrollment for Phase 1 started in August 2022. Phase 2 enrollment and data collection are planned to start in January 2023. Data analysis and results interpretation will take place while patient enrollment and data collection are still ongoing. Details about data collection timeline are provided in Table 4. The last 6 months will be dedicated to manuscript and thesis writing, preparation of final reports, and dissemination of knowledge. Collaborative meetings will be held throughout the entire study period.

Table 4 Participant timeline.

	Study period	
	Enrollment	Allocation	Post-allocation	Close out	
Timepoint	−t1	0	t1	t2	
			Phase 1	Phase 2		
Enrolment:
Screen NICU for eligibility
Approach for informed consent
Study ID created						
X					
X					
	X				
Data acquisition – Phillips Intellivue MX450 Electrocardiogram (ECG)

Photoplethysmography (PPG)

Oxygen saturation (SpO2)

Respiration rate (RR)

– Hourly axial temperature taken by nurses

– ANNE™ Monitoring System Electrocardiogram (ECG)

Photoplethysmography (PPG)

Oxygen Saturation (SpO2)

Respiration Rate (RR)

Chest temperature

Limb temperature

3-axis accelerometry

– Respitrace

– Evaluation of skin at site of device placement.

– Hourly completion of data collection forms by research team

					
				
		X	X		
		X	X		
		X	X		
		X	X		
		X	X		
					
		X	X		
		X	X		
		X	X		
		X	X		
		X	X		
		X	X		
		X	X		
			X		
		X	X		
		X	X		
Assessments
Healthcare worker survey
Parent/guardian survey					
			X	
			X	
Note:

–t1 prior to participation in study, 0—following consent to participate, before data acquisition begins, t1—data acquisition (either for 8 h, 96 h), t2—End of data acquisition, preceding end of study participation.

Sample size

A convenience sample size of 48 neonates was determined, 24 infants for each phase. However, after data is collected from the first 10 patients, an interim analysis will be performed to determine the precision of measurements between the Phillips and the ANNE® One sensors for each of the parameters. Following that, a reassessment of the appropriate sample size will be performed based on this analysis.

Recruitment

NICU personnel will be informed about the research project prior to its initiation and given the opportunity to attend information and trainings sessions to promote collaboration and adherence. Upon study initiation, investigators began to check the NICU for eligible participants (see Fig. 1). Once a potential participant is identified members of the research team approach parents or legal guardians to explain the study and obtain informed consent (see Additional File 3).

Data collection and management

To standardize device administration, all members of the research team will receive training as to how to place devices and connect to study monitor. All data recorded from the different monitoring devices—ANNE® system and standard devices—will be displayed and logged using the BioDASh system. The organization of the data recording framework is shown in Fig. 4. As previously explained, this system provides for the synchronization and aggregation of all data into a single platform for later offline analysis.

Figure 4 BioDASh system architecture and components.

ECG, electrocardiogram; PPG, photoplethysmography; BLE, bluetooth low energy; RIP, respiratory inductance plethysmography; BioDASH, biosensors data aggregation and synchronization system.

The BioDASh application is designed not to interfere with the patient monitor and will not save any protected health information. The streamed data logged by the application will be stored locally on the study laptop with no modifications, post-processing, or filtering. The study laptop will be kept securely with the research team and results will only be shared with team members. Additionally, the logged data will be uploaded to a password protected Dropbox document management software (Dropbox Inc, San Francisco, CA, USA) to which only members of the research team will have access.

Device data collection forms

Two predefined data collection (DCF) forms will be used and are included as Supplemental Material (Additional File 4). The forms will provide specification of device placement, infant body position, concurrent therapeutic treatments, room temperature & humidity (RH32-C2; Omega Engineering, Norwalk, CT, USA), hourly axillary temperature measured by bedside nurse and other details which may be relevant when analyzing the data.

These DCF also include a section for reporting any adverse events such as skin irritation, erosion, or bleeding occurring during participation in the study.

Skin photographs

The de-identified photographs taken will be uploaded to the password protected team Dropbox account at the end of the data collection of each patient.

Survey data

The anonymous responses to the parent and bedside nurse survey will be transcribed throughout the study into an excel file to which only members of the research team will be granted access to.

Statistical methods

Objective 1: Demonstrate the feasibility of continuous wireless monitoring in term and preterm infants with variable degrees of maturation and acuity in the NICU

The number and frequency of the annotations will be used quantitatively to assess challenges related to use of the sensors. Researchers will also report any challenges encountered with the sensor’s battery life, charge duration, and battery durability.

Additionally, we will assess the mean satisfaction scores from nurses and parents as obtained from the surveys. Through review of limited research related to patient satisfaction with ICU monitoring technology, and discussion with the research team a pre-set minim acceptable threshold of 7 out of 10 was set (Poncette et al., 2020; Poncette et al., 2019). This will be used to evaluate the responses by comparing results obtained to pre-set minimum using one sample Wilcoxon test. We will also assess the number of nurses and parents reporting problems with the sensors and identify potential recurring issues as areas for improvement. Descriptive content analysis will be used to code raw responses into categories regarding positive or negative aspects of user experience with the sensors, and the categories and the frequency with which they appear in the surveys will be reported. This analysis and coding of the responses will be accomplished by a member of the research team and a neonatologist.

Objective 2: Assess safety of using a special wireless monitoring system in neonates

A dermatologist will score photographs of the patient skin prior to device placement, and following device removal, using Neonatal Skin Condition Score (NSCS) (Lund & Osborne, 2004). Before and after photograph pairs will be randomized prior to evaluation NSCS scores can range from 3 = intact condition skin to 9 = very poor condition skin. Any change of score from baseline to device removal ( ΔNSCS=NSCSremoval−NSCSbaseline) will be reported as part of the sensor safety analysis. A one sample Wilcoxon test will be performed to determine if the mean change NSCS scores ( ΔNSCS¯)are statistically different than 0 indicating no change in score from baseline.

Safety will also be assessed by evaluating the subjective pain scores obtained with the NIPS results. The average score will be reported, and any score of 3 or greater will be counted as indicative of pain. The total number of scores of 3 or more will be reported. Additionally, a one sample Wilcoxon test will be performed to determine if the average obtained NIPS score is below 3.

Objective 3: Evaluate the accuracy of proposed wireless technology as compared to standard monitoring technology in the NICU

To evaluate the accuracy of the ANNE® we will compare the data obtained by the wireless system to the standard of care. The number of data points acquired for comparison during each hour of participation in either phase 1 or 2 is shown in Table 5. Total number of data points acquired per hour and throughout the entire study are provided in Tables S2 and S3. To comprehensively describe how ANNE® compares to the Phillips monitor, several signal processing and statistical methods have been selected following an extensive literature review. These techniques have been separated into four classes, according to their overall objective: (1) methods that describe signal quality (2) methods that describe the continuity of the data, (3) methods that compare the raw signals from ANNE® and Phillips directly, and (4) methods that compare summary statistics derived from these signals. A list of the statistical methods that will be utilized are outlined in Table 6.

Table 5 Expected number of data points obtained per participant per hour.

	Philips intellivue MX450	Wireless ANNE™ monitoring system	
ECG	1.8 × 106	9.207161125 × 105	
PPG	4.5 × 105	2.303262956 × 105	
Resp signal	2.25 × 105	1.152 × 105	
HR	3.515625 × 103	3.6 × 103	
SpO2	3.515625 × 103	3.6 × 103	
RR	3.515625 × 103	3.6 × 103	
Temperature (axial & chest respectively)*	1	9 × 102	
Skin temp. limb	N/A	9 × 102	
3-axis Accel.	N/A	1.5 × 106	
Total	2.485547875 × 106	2.778842408 × 106	
Note:

* Temperature is recorded manually by nurse every hour.

Table 6 Analytical methods to be implemented.

Measure	Applicable methods	
Signal quality	Time-warped polynomial filtering (TWPF)
Signal averaging
Probability distribution function (PDF)
Power spectrum	
Continuity of the signal	FIR antialiasing lowpass filtering
Movement artifact removal
Data segmentation	
Direct comparison of the raw signals	Cross-correlation function
Probability distribution function (PDF)
Power spectrum
Impulse response function (IRF)
Box-jenkins system identification
Bland-altman method
Lin’s concordance correlation coefficient (CCC)
Intraclass correlation coefficient (ICC)
Paired t-test
Equivalence testing	
Comparison of derived statistics/signals	Lin’s concordance correlation coefficient (CCC)
Intraclass correlation coefficient (ICC)
Paired t-test
Equivalence testing
ANOVA F-test	

Table 5 Expected number of data points obtained per participant per hour.

Quality of the signals—by quantifying the noise level of each device. This will be accomplished via signal averaging and time-warped polynomial filtering (TWPF). Both methods attempt to clean a noisy signal such as an ECG and produce a deterministic product. By removing this deterministic component from the raw signal, a noise signal may be obtained for each device. Statistical parameters including variability metrics and the signal-to-noise ratio (SNR) will be used to describe the overall quality of each device for measuring vital signals. These parameters will be accompanied by appropriate probability density functions (PDF) and power spectra.

Continuity of the signals—pre-processing will be used to divide the data into segments of connection loss, corruption due to significant movement artifact, and usable portions of signal recordings. From here, a set of statistics can be used to describe the technical reliability of ANNE®, including the number and duration of interruptions, the average period between consecutive interruptions, as well as the portion of data deemed unreliable due to increased movement. Small segments of interruption (<2 s) will be interpolated to yield longer segments of continuous monitoring. Performing this analysis on the data acquired from both phases will further allow us to study how increasing recording length potentially exacerbates these statistics.

Compare the raw signals from ANNE® and Phillips directly—a preliminary step requires that the signals from ANNE® be resampled to produce segments with an equal number of data points as the Philips monitor (this will be completed as part of the pre-processing in the second class of analysis). Any recording period with under 2 h of usable paired data from the ANNE® and the Phillips will be excluded from comparative analysis. Once equal data samples have been obtained from all usable recordings, the cross-correlation will be used to quantify the latency between matching signals. Performing this operation on segmented epochs throughout a recording will also allow us to determine whether the delay varies with monitoring time or remains constant. The length of epoch segments will be determined based on preliminary analysis. In the time domain, the (PDF) will be used to compare the amplitude structures of signals, while system identification via linear models will be used to study the relationship between ANNE® and the Philips monitor. In the frequency domain, the power spectrum will be derived to compare the frequency contents of each signal and extract statistics for the third class of techniques. Lastly, statistical methods, such as the Bland-Altman method, paired t-tests, and correlation coefficients, will be implemented to evaluate the agreement between paired measurements or samples.

Compare summary statistics—from the raw signals, summary statistics will be derived and compared to further validate the reliability of ANNE®. A detailed list of statistics, both in the time domain and frequency domain, for the various vital signals are outlined in Table 3 alongside traditional measures such as the average heart rate over a specified period, the average respiratory rate over a specified period, and the number of heart beats or breaths detected and/or missed by each device. Graphical representations of desired statistics used for analysis are shown in Fig. 5. The same statistical procedures previously mentioned will be applied once more, this time comparing paired statistics as opposed to matching sample points.

Figure 5 Sample of derived comparative statistics for heart rate (HR).

(A) Detrended (via removing the mean) tachogram representing overlayed heart rate values obtained from Phillips (blue) and ANNE (red) over 4,000 s window (B) probability density function/histogram representing ratio of sample points from each device corresponding to heart rate value computed from tachogram for each respective device over 4,000 s window, Phillips (blue) and ANNE (red) (C) power spectrum representing frequency distribution of the tachogram signals over 4,000 s window for each device Phillips (blue) and ANNE (red).

The above methods have become widely accepted means for evaluating monitor performance in the medical industry and have set a standard for assessing novel devices before deployment. Using MATLAB, we can expect the analysis to be both expressive and efficient, yielding abundant metrics to characterize the reliability and accuracy of ANNE®.

Enhanced respiratory monitoring using accelerometry

For patients in Phase 2, we will compare the chest and abdominal wall movement signals derived from accelerometry with RIP (the gold standard) and thoracic impedance (the current standard) with the simultaneously recorded wireless wearable data.

Data monitoring and harms

The wireless devices utilized have been piloted before in NICU babies and are designed to adhere more gently to the skin than the adhesives used in standard of care, and thus will not cause additional risk or discomfort. Further, participation in this study will not interfere with the standard of care received. Patient participation can be stopped at any point if parents desire or upon decision of the attending physician (ex: in case of clinical deterioration). In the event of an adverse event such as skin erosion, bruising, bleeding or other it will be reported in the DCF as well as submitted to the McGill University Health Center’s Research Ethics Board to respect the protocol and Good Clinical Practice.

Confidentiality

All recorded data will be kept at the MUHC and will not be shared with any external collaborators. All identifying information will only be shared with members from the study team and will be omitted from any publication or documents shared with individuals external to research team.

The study will be overseen by the principal investigators. They will meet periodically to assess its progress and provide scientific direction. They will also be responsible for protocol execution, routine monitoring of data quality, recruitment, and retention, and that the study is meeting key milestones consistent with the timeline.

Access to data

De-identified data and metadata collected during this study will be submitted to an appropriate public repository at the time the results are published.

Discussion

Current vital signs monitoring technology in the NICU is indispensable but uses wires and cables. The wires may interfere with routine care, cause discomfort to patient, pose risk of infection, and act as a barrier to parent-infant contact. New advances in wearable technology, if properly applied to the NICU context, have the potential to facilitate patient care and provide acquisition of new signals. Thus, the development and testing of such technologies is warranted.

This proposed study will build on previous pilot using the ANNE® One in the NICU context (Russotto et al., 2015; Chen et al., 2009). The system will be evaluated using a larger sample size including a wider range of gestational age (extremely premature infants) and a diversity of clinical conditions. Further, re-evaluation of safety and efficacy of this technology are an essential step towards the implementation of these wireless monitoring systems as standard of care. Finally, this will be the first study using continuous wireless monitoring technology in the NICU in Canada.

The study has some limitations. This is an observational study with a convenience sample. However, for the 2 phases proposed, the aim of the study is to explore the feasibility of wireless monitoring technology in the NICU, and therefore its findings could be used as the basis for a future larger study with longer monitoring durations. Another anticipated limitation is the possibility of incomplete data collection. There is a chance that a recruited patient may be discharged or transferred during the study and data collection discontinued prematurely. However, based on average duration of stay of patients in the NICU this should not interfere with sufficient patient data collection.

Supplemental Information

Supplemental Information 1 Standard Protocol Items Recommendations for Interventional Trials (SPIRIT) checklist for a study protocol.

List of necessary components to meet SPIRIT guidelines and their locations (i.e., page number) in the manuscript.

Click here for additional data file.

Supplemental Information 2 Surveys provided to parents and nurses caring for study participant.

Click here for additional data file.

Supplemental Information 3 Data collection form to be completed for each participant in the study.

Click here for additional data file.

Supplemental Information 4 Data annotations index that will be used to add annotations into BioDASH recording system during study.

List of abbreviations used to annotate different types of clinical events and care which may occur during study monitoring periods and impact data quality and values.

Click here for additional data file.

Supplemental Information 5 Expected number of data points obtained per participant in phase 1 and 2 of study.

Number of data points obtained per participant for each signal monitored based on duration of monitoring and sampling rate of monitoring technology.

Click here for additional data file.

Supplemental Information 6 Expected total number of data points acquired in each phase of study.

Total number of data points obtained across all participants participant for each signal monitored based on duration of monitoring and sampling rate of monitoring technology.

Click here for additional data file.

We would like to acknowledge the collaboration with iKinesia Inc for the development of the BioDASh system to this study.

Additional Information and Declarations

Competing Interests

Author Contributions

Ethics

Data Availability

The authors declare that they have no competing interests. Sibel Health was not involved in the study design and will not be involved on data collection and analysis.

Eva Senechal conceived and designed the experiments, prepared figures and/or tables, authored or reviewed drafts of the article, and approved the final draft.

Daniel Radeschi conceived and designed the experiments, prepared figures and/or tables, authored or reviewed drafts of the article, and approved the final draft.

Lydia Tao conceived and designed the experiments, authored or reviewed drafts of the article, and approved the final draft.

Shasha Lv conceived and designed the experiments, authored or reviewed drafts of the article, and approved the final draft.

Emily Jeanne conceived and designed the experiments, authored or reviewed drafts of the article, and approved the final draft.

Robert Kearney conceived and designed the experiments, prepared figures and/or tables, authored or reviewed drafts of the article, and approved the final draft.

Wissam Shalish conceived and designed the experiments, prepared figures and/or tables, authored or reviewed drafts of the article, and approved the final draft.

Guilherme Sant Anna conceived and designed the experiments, prepared figures and/or tables, authored or reviewed drafts of the article, and approved the final draft.

The following information was supplied relating to ethical approvals (i.e., approving body and any reference numbers):

This study has been reviewed and approved by the Pediatric (PED) panel of the Research Ethics Board of the McGill University Health Centre (#2022-7704).

The following information was supplied regarding data availability:

This is a registered report (i.e., a study protocol) so there is no data reported in this article.

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
