# Peer review of "The use of wireless sensors in the neonatal intensive care unit: a study protocol"

_PeerJ, doi:10.7717/peerj.15578_

## Round 0.1 · original submission · Minor Revisions

It needs minor revision before acceptance.

Reviewer 1 ·

Basic reporting

1. Authors well described the current clinical workflow and practice in details as well as why a new innovation needs to be validated to address the clinical needs.
2. Authors also successfully outlined commercial options available with their limitations to justify the needs of this study with ANNE System.
3. The system seems highly adequate to fit into this study’s goal.
4. The study is well designed with the wide range of age groups, especially group 2 that includes preterm infants to understand the real feasibility of the system.
5. The study is well designed in terms of risk assessment, gathering feedback by nurses, and evaluation of condition of babies after study such as skin evaluation.
6. The study is also well designed to test real-life accuracy of the ANNE System as a comparison to standard monitor technology as a gold standard.
Overall, the paper is well designed and written without needing further justification.

Experimental design

See answer to question 1

Validity of the findings

See answer to question 1

·

Basic reporting

Lines 56-57
"Heart rate is measured via small electrodes on the bedside monitor that continuously displays both electrocardiogram (ECG) tracings and average HR."
Sentence readability can be improved. The electrodes are connected via wires to the bedside monitor.

Line 59 (and others throughout)
The "2" of SpO2 should be sub-script.

Lines 66-67
A reference for the impact on parental mental health should be included.

Lines 69-71
Reference 8 does not support this statement in full. It discusses the reported tangling of wires and the impact on handling and movement of babies. A reference for the "increased risk of nosocomial infections" is required.

Lines 109 & 344
What does "acuity" mean in this context?

Line 111
Replace first "and" with a comma.

Line 172
"Participation in study..."
Insert "the" before "in".

Lines 188-189
A pulse oximeter uses two LEDs at different wavelengths, not one. Furthermore, the SpO2 value is derived from PPG signal at each wavelength. So, it would be correct to say “…, and two LEDs along with a photodiode that captures the photoplethysmogram (PPG) at each wavelength of light, from which the SpO2 is calculated.”.

Experimental design

Given many neonates have their temperature continuously monitored, what is the rationale for not including continuous temperature monitoring in this study to compare with the continuous temperature monitoring of the Anne One system?

Line 204
In the standardised list of annotations, additional information could be gathered on the reason for sensor adjustment or replacement. So, for example, if the sensor is changed due to low battery, or if the skin adhesion has become loose. This would aid evaluating the usability of the system. I note that line 347 mentions this, but can more detail be given on how this will be collected.

What is the reason for not using the respiratory monitoring that is built into the Philips bedside patient monitor (impedance pneumography through the ECG electrodes)? It would be preferable to use minimal additional equipment where possible, and it would simplify the experimental setup. Furthermore, it would enable continuous monitoring and not for a sub-sample of the time period.

Lines 255 & 437
Due to the additional weight (versus conventional sensors) that the wireless sensors present, has any thought been given to how the potential impact of this could be assessed? Perhaps a specific question in the qualitative feedback survey could ask about this. This is particularly relevant for the extremely pre-term infants that the study hopes to recruit.

Have parents been involved in the study design? If so, can their involvement be mentioned. In particular, a cohort of parents should have had an opportunity to review the participant information sheet and the qualitative questions that will be asked.

Line 309-310
Is it only members of the research team that will place the devices? Given this study is testing the feasibility of the device, should it not be that non-research nursing staff should be using the devices (with appropriate training)?

Line 350
How has the threshold of 7 out of 10 been determined?

Figure 1
What is the rationale for exclusions other than skin-related reasons? Does the device’s intended use particularly exclude such conditions, or is this imposed for other reasons?
“Not approached/missed” should be in the top box, not the second.

Is it planned that the devices be used for participants at the same time? Is so, has the risk of transmission to the wrong tablet device been assessed? Similarly, are co-bedded patients (e.g., twins) included in the study? If so, will the Anne One device be able to be used on both at the same time? This would be a useful test of the device’s pairing strategy and security, if possible.

Can the authors include a statement about infection control? In particular, what the regime is for cleaning the reusable parts between patients.

Table 1
Can the authors clarify if the ages listed are the age at birth or the age at enrolment?

Table 4
I believe there has been a typo under Philips MX450, ECG. An “X” should be in both t1 columns.

Validity of the findings

No comment.

Additional comments

This is a well written protocol with thorough data collection and analysis included. The comments above are mostly minor, although particular attention is drawn to feedback on the weight of the devices, parental involvement in study design, non-research staff involvement in sensor use, exclusion criteria, and sensor mis-pairing risks.

I also note that the relationship with Sibel Healthcare has not been declared. The attached SPIRIT checklist indicates that a declaration of interests has been included, but I cannot find such a declaration. What involvement has Sibel had in the study design or will have in the execution? Is data from the study (other than that publicly available on publication) to be shared with Sibel?

---

## Round 0.2 · accepted · Accept

Satisfactory revised. Congratulations!

·

Basic reporting

No comment

Experimental design

I would urge the authors to include parents and families in the actual design of future research (co-production), not just in the data gathering. It is my experience that this process is hugely beneficial for improving the study design, increasing recruitment rates, and developing a better relationship between researchers and the parent community, ultimately leading to better research outcomes.

Validity of the findings

No comment

Additional comments

I thank the authors for their considered response to all the feedback points I raised. I am satisfied that these have been resolved in full, and I therefore recommend this article be accepted with no further revisions. Best of luck with recruitment!